# Suppression Head Impulse Test (SHIMP) versus Head Impulse Test (HIMP) When Diagnosing Bilateral Vestibulopathy

**DOI:** 10.3390/jcm11092444

**Published:** 2022-04-26

**Authors:** Tessa van Dooren, Dmitrii Starkov, Florence Lucieer, Bieke Dobbels, Miranda Janssen, Nils Guinand, Angelica Pérez Fornos, Herman Kingma, Vincent Van Rompaey, Raymond van de Berg

**Affiliations:** 1Division of Balance Disorders, Department of Otorhinolaryngology and Head and Neck Surgery, Maastricht University Medical Centre, 6229 HX Maastricht, The Netherlands; f.lucieer@gmail.com (F.L.); nils.guinand@hcuge.ch (N.G.); hermanuskingma@gmail.com (H.K.); raymond.vande.berg@mumc.nl (R.v.d.B.); 2Faculty of Physics, Tomsk State Research University, 634050 Tomsk, Russia; dstark2048@gmail.com; 3Faculty of Medicine and Health Sciences, University of Antwerp, 2000 Antwerp, Belgium; biekedobbels@gmail.com (B.D.); vincent.vanrompaey@uantwerpen.be (V.V.R.); 4Department of Otorhinolaryngology and Head and Neck Surgery, Antwerp University Hospital, 2650 Edegem, Belgium; 5Department of ENT/Audiology, School for Mental Health and Neuroscience (MHENS), Maastricht University Medical Centre, 6229 HX Maastricht, The Netherlands; miranda.janssen@maastrichtuniversity.nl; 6Department of Methodology and Statistics, Care and Public Health Research Institute (CAPHRI), Maastricht University, 6211 LK Maastricht, The Netherlands; 7Service of Otorhinolaryngology Head and Neck Surgery, Department of Clinical Neurosciences, Geneva University Hospitals, 1205 Geneva, Switzerland; angelica.perez-fornos@hcuge.ch

**Keywords:** SHIMP, HIMP, VHIT, covert saccades, compensatory saccades, VOR gain, bilateral vestibulopathy

## Abstract

The Suppression Head Impulse (SHIMP) test was introduced as an alternative to the Head Impulse Paradigm (HIMP) to overcome challenges in VOR gain calculation due to the interference of covert saccades. The objectives of this study were (1) to determine if SHIMP, compared to HIMP, reduces covert saccades in BV patients and (2) to define the agreement on diagnosing BV between SHIMP and HIMP. First, the number of covert saccades was compared between SHIMP and HIMP. Secondly, VOR gain was compared between SHIMP and HIMP. Lastly, the agreement between SHIMP and HIMP on identifying BV (horizontal VOR gain <0.6) was evaluated. A total of 98 BV patients were included. To our knowledge, this is the largest study population on SHIMP testing in BV patients. Covert saccades were significantly reduced, and a lower VOR gain was found during SHIMP compared to HIMP (*p* < 0.001). However, the clinical relevance of these statistically significant differences is small. In 93% of the patients, an agreement was found between the two paradigms regarding the diagnosis of BV, and both paradigms detect BV in the vast majority of patients.

## 1. Introduction

The Head Impulse test (HIMP), first described in 1988, is nowadays widely used to assess the vestibulo-ocular reflex (VOR) function of all semicircular canals in the high-frequency domain [1]. During this test, the examiner performs fast head impulses (>120°/s) and passive head movements with a small amplitude (10–30°), unpredictable in timing and direction. Subjects are asked to fixate on an earth-fixed target at eye level in front of them. In the case of a normal VOR, the eyes will immediately move in the opposite direction of the head impulse to assure gaze stability on the target. In patients with a deficient VOR, the eyes will move slower than the head or even initially move along with the head. To correct for the loss of gaze fixation, a compensatory eye movement (saccade) is required. Therefore, the appearance of saccades could indicate vestibular hypofunction. These saccades can appear after (i.e., overt saccade) or during (i.e., covert saccade) the head impulse. Overt saccades are often detected by the naked eye of the examiner. In contrast, this is mostly impossible for covert saccades [2].

The HIMP can also be performed using a device that allows for the quantification of the VOR and detection of all compensatory saccades: the video head impulse test (VHIT). This device tracks head and eye movements during the head impulse test. Different types of devices are commercially available, including systems with head-mounted lightweight goggles or an earth-fixed remote camera. The main outcome parameter is VOR gain, calculated as the ratio between eye and head movement. VOR gain will be close to one in healthy subjects and lower in patients with a deficient VOR [3]. For example, a bilateral horizontal VOR gain of <0.6 is one of the main criteria for the diagnosis of bilateral vestibulopathy (BV) [4]. Different algorithms can be used to calculate VOR gain. Covert saccades might challenge VOR gain calculation due to their interference with eye movements produced by the VOR [5]. This implies that VOR gain might not always perfectly reflect the VOR function. Current HIMP systems tend to overcome this issue by, for example, desaccading eye movements [6].

In 2016 the Suppression Head Impulse test (SHIMP) was introduced by MacDougall et al. as an alternative to HIMP to overcome challenges in VOR gain calculation due to the interference of covert saccades [7]. The main difference between SHIMP and HIMP is a head-fixed target instead of an earth-fixed target. The target is a laser dot projected by lightweight goggles. As a result, the target moves along with the head during the head impulse. In the case of an adequate VOR, the eyes will initially move in the opposite direction of the head. However, since the head-fixed target has moved during the impulse, these subjects need compensatory eye movements (saccades) to bring the eyes back on the target. Consequently, saccades during SHIMP represent (residual) vestibular function, while saccades during HIMP could indicate a vestibular loss [7]. Moreover, saccades in SHIMP testing will mainly occur after the head impulse (overt saccades) and not during the head impulse (covert saccades) [2]. However, studies show that predictability during SHIMP could still result in shorter latency of saccades and even covert saccades [8]. Hence, when properly performed, SHIMP testing could enable elimination over covert saccades and might facilitate a more precise VOR gain calculation than in HIMP.

Previous research demonstrated that SHIMP is a feasible test in healthy subjects and vestibular patients. In SHIMP, a lower VOR gain was found compared to HIMP. The underlying mechanism is not fully known, but several explanatory theories are preferred: less interference of covert saccades as described above (no desaccading of the traces necessary) or the influence of compensatory mechanisms that are possible during SHIMP (e.g., VOR cancellation/inhibition resulting in slower eye velocities) [7,9,10]. The presence of covert saccades is lower in SHIMP than in HIMP [7]. However, the clinical consequence of eliminating covert saccades when using SHIMP has not yet been determined comprehensively in a large group of BV patients. 

Therefore, the objectives of this study were (1) to determine if SHIMP, compared to HIMP, reduces covert saccades in BV patients and (2) to define the agreement on diagnosing BV between SHIMP and HIMP. It was hypothesized that BV patients demonstrated fewer covert saccades and a lower VOR gain when tested with SHIMP compared to HIMP, but that these effects might not influence the diagnosis of BV in most patients.

## 2. Methods

### 2.1. Study Population

This study comprised patients diagnosed with BV at the Division of Balance Disorders at Maastricht University Hospital in the Netherlands and Antwerp University Hospital in Belgium, based on the diagnostic criteria for BV from the Bárány Society [4]. Inclusion criteria comprised (1) reduced caloric response (sum of bithermal maximum peak slow phase eye velocities of <6°/s on each side), (2) and/or reduced horizontal angular VOR gain < 0.1 on a rotatory chair and a phase lead >68°, (3) and/or bilateral horizontal VOR gain < 0.6, obtained by the VHIT. Exclusion criteria comprised being unable to stop vestibular suppressants for one week (cinnarizine and all psychiatric medication) and the inability to undergo one of the vestibular examinations.

#### Study Design

A systematic approach was used. First, it was determined whether covert saccades were eliminated during SHIMP by comparing the number of covert saccades between SHIMP and HIMP. A custom-made algorithm detected saccades after strict trace evaluation to exclude artefacts as described in Section 2.3). Since the definition of covert saccades can be different between clinics, the latency of the first saccade (covert and/or overt) was also analyzed separately. Secondly, the VOR gain was compared between the two paradigms, and the influence of peak head velocity was determined. Lastly, the agreement between SHIMP and HIMP on identifying BV according to the diagnostic criteria (horizontal VOR gain <0.6) was evaluated. For this last analysis, the unfiltered data from the device were used, as will be the case in daily practice.

### 2.2. Experimental Setup

To reduce the artefacts to a minimum, two trained examiners (FL, BD) followed a strict experimental setup, as described in previous articles [11,12]. Every patient underwent testing in the same order (first HIMP, then SHIMP). All tests were performed using the ICS Impulse system (Natus, California, CA, USA). Distance to the target and room illumination were similar for all patients [13]. The right eye was tested in both SHIMP and HIMP paradigms. After calibration, the examiner nor the patient were allowed to touch the strap and the goggles. Patients were constantly kept alert by the instructions of the examiner. Fast (>120°/s), outwards, horizontal head impulses with a small amplitude (10–30°) were given, unpredictable in timing and direction [14,15].

### 2.3. Saccades

#### 2.3.1. Saccade Detection 

In order to determine saccades, first head and eye velocity traces were exported from the Otometrics system, and position and acceleration data were calculated using Wolfram Mathematica 11.3 (Wolfram Research, Champaign, IL, USA) [16]. Only traces that were accepted by the Otometrics system itself were exported. All traces were checked on artefacts. Traces were excluded from analyses when (1) peak head velocity was <120°/s, (2) the head velocity trace had a bounce >50% of peak head velocity after the head impulse, (3) head velocity never crossed zero after peak head velocity, (4) the head velocity trace contained missing values, (5) the head velocity trace differed from the standard shape, assessed by visual inspection and consensus between three authors (RB, DS, TD), or (6) when the mean head velocity of the interval of 80 ms prior and 120 ms after a peak head velocity was not in the range of ± 3 SD of the set of mean head velocities calculated in the same interval in all traces of one patient [17].

A custom-made algorithm was applied to extract saccades from the eye acceleration traces, yet every saccade was verified by visual inspection of the velocity and position traces. Two authors needed to achieve consensus (TD, DS) before a saccade was approved. Saccades were included when they (1) occurred after peak head velocity, (2) had a magnitude of more than 60°/s, and (3) the peak saccade velocity was recorded. The onset of a saccade was the point where eye velocity crossed zero or eye acceleration reached 2000°/s^2^. The offset of a saccade was the point where eye acceleration crossed zero after eye velocity crossed zero, or acceleration was below 2000°/s^2^ when velocity did not cross zero. A saccade was classified as “covert saccade” when onset occurred before head velocity crossed zero and as “overt saccade” when onset occurred after head velocity crossed zero. Head impulse onset was set on head velocity exceeding 10°/s. Head impulse offset was defined as head velocity crossing zero (Figure 1).

#### 2.3.2. Presence of Covert Saccades

The presence of covert saccades for every patient was determined as the frequency of occurrence of at least one covert saccade per trial. Every trial consisted of seven artefact-free traces (as described above) [18]. Only the first saccade of a trace was used for analysis. As a result, every patient had a minimum of zero and a maximum of seven covert saccades per trial. The frequency of occurrence of a covert saccade was first registered as a binary outcome (yes/no) for every trace separately. From these data, a ratio (0–1) and percentage (0–100%) per patient were calculated.

#### 2.3.3. Latency of Saccades

The latency of the first saccades was extracted from the original eye velocities in the Otometrics system. Both overt and covert saccades were included. Latency (in milliseconds) was registered as the onset of the saccade and was normalized to the start of the head impulse.

### 2.4. VOR Gain 

For both HIMP and SHIMP, VOR gain was calculated by the Otometrics system itself over all traces accepted by the system. VOR gain was also calculated with a custom-made algorithm, using the raw data extracted from the Otometrics system. This VOR gain was calculated over the first seven artefact-free traces of every patient. Both methods (Otometrics system and custom-made algorithm) calculated the VOR gain by the ratio of the area under the curve of eye movement and head movement. The eye movement was desaccaded if needed [6]. To detect influences of head velocity on VOR gain outcomes in this study, peak head velocities were compared between HIMP and SHIMP. 

### 2.5. Statistical Analysis

Data were analyzed using SPSS Statistics 25 (IBM, Armonk, NY, USA) for Windows and R (v.3.5.2.). The α-value was set at *p* < 0.05. 

#### 2.5.1. Statistical Analysis of Saccades

##### Covert Saccades

Marginal multilevel model analysis was applied with side (right/left) and test (HIMP/SHIMP) as independent variables and an unstructured covariance matrix of the residuals to detect a statistically significant difference in the frequency of covert saccades (ratio 0–1) in BV patients between HIMP and SHIMP testing. 

##### Latency of First Saccade (Covert and/or Overt)

A two-sided paired t-test was used to compare the latency (ms) of the first saccade between HIMP and SHIMP. This analysis included the first saccade (i.e., both covert and overt saccades) of the first seven artefact-free traces in every patient. Logically, patients without a saccade in HIMP or SHIMP were not included in this part of the analysis. 

#### 2.5.2. Statistical Analysis of VOR Gain 

Marginal multilevel linear regression with side (right/left), VOR gain, and test (HIMP/SHIMP) as independent variables and an unstructured covariance matrix of the residuals were performed to detect a statistically significant difference in VOR gain in BV patients between HIMP and SHIMP testing. VOR gain as calculated by a custom-made algorithm over the first seven artefact-free impulses was used for analysis. 

#### 2.5.3. Statistical Analysis of Peak Head Velocity

The difference in peak head velocities between HIMP and SHIMP was calculated with a two-sided paired t-test. Median peak head velocities (extracted from the raw traces of the VHIT system) of the traces used to calculate VOR gain were used for analysis. 

#### 2.5.4. Analysis of Agreement between HIMP and SHIMP Regarding BV Diagnosis

For this analysis, patients were excluded if diagnosed with BV solely based on VHIT outcomes since VOR gain obtained by the VHIT was used as the outcome parameter. VOR gain calculated by the Otometrics system (using all accepted traces) was used, as will be the case in daily practice. A HIMP VOR gain of <0.6 was classified as “bilateral vestibulopathy”, and a VOR gain of ≥0.6 was classified as “no bilateral vestibulopathy” [4]. For SHIMP, two different cut-off values (<0.6 and <0.5) were used and separately analyzed. In case the paradigms (HIMP and SHIMP) showed a discrepancy in classifying BV, the patient was classified as “no agreement”. In patients with “no agreement”, visual inspection and descriptive analysis by two authors (TD, RB) were performed. This comprised inspecting the presence and timing of covert saccades, comparing VOR gain calculated by the system and the custom-made algorithm, and assessing if the traces showed characteristics of BV. 

### 2.6. Ethical Considerations

This study was conducted following the legislation and ethical standards on human experimentation in the Netherlands and the Declaration of Helsinki (amended version 2013). Approval was obtained from the ethical committee of Maastricht University Medical Center (NL52768.068.15/METC). All subjects provided written informed consent.

## 3. Results

### 3.1. Patient Characteristics

The study population comprised 98 BV patients from the Netherlands and Belgium: 56 males and 42 females. Mean age was 59 years old (SD 13 years). Definite and probable etiologies included: ototoxic effects of antibiotics (*n* = 12) or chemotherapy (*n* = 2); post-infectious due to Lyme disease (*n* = 2), cerebral malaria infection (*n* = 1), herpes infection (*n* = 1), meningitis (*n* = 6), or neuritis (*n* = 4); head trauma (*n* = 5); inherited by DFNA9 gene mutation (*n* = 13) or other gene mutations (*n* = 10); bilateral Menière’s disease (*n* = 6); autoimmune disease (*n* = 2). In 34 patients, no etiology could be determined (idiopathic). 

A representative sample of eye and head movements obtained with HIMP and SHIMP is illustrated in Figure 2. Further details of the VHIT characteristics (saccades, VOR gain, and peak head velocity) of all tested patients will be discussed below.

### 3.2. HIMP versus SHIMP: Presence of Covert Saccades

A statistically significant difference was found in the presence of covert saccades between SHIMP and HIMP (F(1,97) = 86.314, *p* < 0.001). During SHIMP testing, fewer covert saccades were produced compared to HIMP (estimated difference SHIMP-HIMP = −0.289 (−0.351, −0.227)). A covert saccade was present in 34–35% of the HIMP traces and 5–6% of the SHIMP traces (Figure 3A).

### 3.3. HIMP versus SHIMP: Latency of the First Saccade (Covert and/or Overt)

This analysis comprised 92 patients for leftwards impulses and 93 patients for rightwards impulses since patients without a saccade in HIMP or SHIMP could not be included. A statistically significant difference was found in the latency of the first saccade between SHIMP and HIMP (*p* < 0.001). Saccades appeared later (i.e., demonstrated a longer latency) during SHIMP testing. The mean latency of the first saccade was 276 ms for leftward head impulses and 274 ms for rightward impulses during SHIMP, and during HIMP, 193 ms for leftwards head impulses and 197 ms for rightwards head impulses (Figure 3B).

### 3.4. HIMP versus SHIMP: VOR Gain Differences

Mean VOR gain in SHIMP was lower compared to HIMP (estimated difference SHIMP-HIMP = −0.026 (−0.040, −0.012). This difference was statistically significant (F(1,97) = 12.913, *p* < 0.001). Mean VOR gains for rightward, and leftwards head impulses were, respectively, 0.32 and 0.33 in SHIMP and 0.35 and 0.35 in HIMP (Figure 3C).

**Figure 3 jcm-11-02444-f003:**
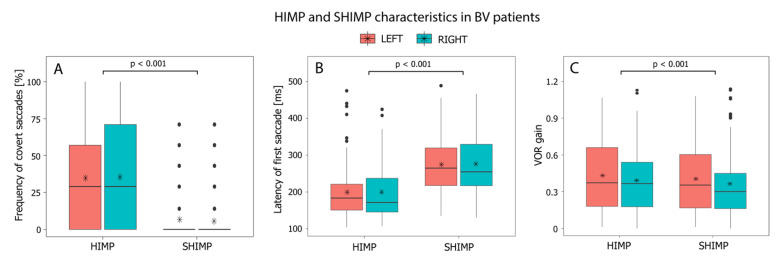
Characteristics of HIMP and SHIMP testing in BV patients for rightwards and leftwards head impulses: frequency of covert saccades (**A**), the latency of first saccade (covert and/or overt), (**B**), and VOR gain as calculated by a custom-made algorithm (**C**). Black horizontal lines represent median values; asterisks represent mean values for all patients.

### 3.5. HIMP versus SHIMP: Peak Head Velocity

Median peak head velocity was significantly lower during SHIMP compared to HIMP (*p* < 0.001) (Appendix A).

### 3.6. Analysis of Agreement between HIMP and SHIMP Regarding BV Diagnosis

Six patients were excluded from this analysis since diagnosis of BV was solely based on VHIT outcomes, as described in paragraph 2.6.4. In 93% of the 92 patients, HIMP and SHIMP agreed on the diagnosis of BV (either “bilateral vestibulopathy” or “no bilateral vestibulopathy”) when using the cut-off value of 0.6 for both paradigms (Table 1). In six patients (7%), the two paradigms did not agree on the diagnosis of BV. All these six patients were classified as “BV” with SHIMP and “no BV” with HIMP. However, in three out of these six patients, HIMP and SHIMP agreed when using the VOR gain calculated by the custom-made algorithm. In the other three patients with no agreement, the visual inspection did show pathological eye responses, but this was not reflected by a VOR gain <0.6. In case a SHIMP cut-off value of <0.5 was used, agreement on the diagnosis of BV increased to 97% (Table 1).

## 4. Discussion

This study compared the outcomes of SHIMP and HIMP in a large group of 98 patients with bilateral vestibulopathy (BV), diagnosed according to the diagnostic criteria of the Báràny Society [4]. To our knowledge, this is the first study to compare SHIMP and HIMP in a patient population of this size. 

SHIMP significantly reduced the number of covert saccades and VOR gain compared to HIMP. More importantly, in 93% of the patients, an agreement was found on the BV diagnosis between the two paradigms. 

### 4.1. HIMP versus SHIMP: Covert Saccades

Significantly fewer covert saccades were produced by BV patients tested with SHIMP compared to HIMP (0.05 vs. 35%) (Figure 3). This “covert saccade killer” phenomenon is in agreement with previous studies on smaller populations of patients with a vestibular deficit [7,9,19]. The elimination of covert saccades should facilitate a more accurate gain calculation [7]. This is especially valuable in a BV population, in which patients often produce covert saccades [5].

### 4.2. HIMP versus SHIMP: VOR Gain

VOR gain in SHIMP was significantly lower than in HIMP. However, the clinical implication of the VOR gain difference is small: only a mean difference of 0.02 (leftwards impulses) and 0.03 (rightwards impulses) (Figure 3). This VOR gain difference between both paradigms is slightly smaller but comparable to previous results in smaller groups of healthy subjects and BV patients [7,9]. The underlying mechanism of a lower VOR gain in SHIMP is not fully known, but several explanatory theories are preferred. For example, the reduction of covert saccades could provide a more precise VOR gain calculation in SHIMP. However, a VOR gain difference (larger than in this BV population) between these paradigms was also demonstrated in studies with healthy subjects (without covert saccades in HIMP testing) [9,20]. This might be explained by VOR response suppression, in which subjects decrease their VOR response. VOR suppression in unexpected passive movements is observed within 60–90 ms after the start of head movement; therefore, it could be reflected in a lower VOR gain during SHIMP testing [10,21]. Furthermore, higher head velocities result in lower VOR gains [22]. In this study, peak head velocities were significantly lower during SHIMP testing, which could therefore not justify the lower VOR gains in SHIMP.

### 4.3. HIMP versus SHIMP: Agreement on the Diagnosis of BV

Agreement between HIMP and SHIMP on the diagnosis of BV (VOR gain < 0.6) was found in 93% of this population (Table 1). This suggests that the significant differences observed between both paradigms (presence of covert saccades and VOR gain) probably have minor clinical consequences since both paradigms detect BV in the vast majority of the patients. 

The six patients in which HIMP and SHIMP did not agree on the BV diagnosis (when using a SHIMP cut-off value of <0.6) were all diagnosed as BV by SHIMP and not with HIMP. These discrepancies could be attributed to gain calculation and cut-off values. Regarding gain calculation, a custom-made algorithm and visual inspection of the traces did show severe vestibular hypofunction in these cases in both paradigms. Although, it must be stressed that also, with the custom-made algorithm, no agreement was found between both paradigms in 5 out of 92 patients. This demonstrates the need for a standardized approach for evaluating and interpreting head impulse testing outcomes. This should include a universal gain calculation algorithm combined with an assessment of the raw traces [5,8]. Regarding cut-off values, two cut-off values were used for SHIMP in this study (VOR gain <0.6 and <0.5). Although no official cut-off values have been published for SHIMP, it was previously proposed to state a lower cut-off value, considering the lower VOR gain values during SHIMP [23]. In this study, lowering the SHIMP cut-off value to 0.5 increased the agreement between HIMP and SHIMP to 97%. However, an increase in the agreement does not imply an increase in the correctly made BV diagnoses. After all, fewer patients were diagnosed with BV after lowering the cut-off value to 0.5, while BV was already demonstrated by caloric testing and/or rotatory chair testing. This implies that future research is needed to determine the proper cut-off value for SHIMP in BV.

### 4.4. HIMP versus SHIMP: The Daily Practice

Both HIMP and SHIMP were well tolerated by all patients, and some of them reported that SHIMP testing felt more like a game than a medical test. Unfortunately, the current clinically available SHIMP software does not include testing of vertical semicircular canals. Therefore, when testing of all six semicircular canals is needed (e.g., in a research setting, such as vestibular implant research), HIMP testing is preferred [24]. Nevertheless, since SHIMP was demonstrated to be a “covert saccade killer”, SHIMP could be an alternative in clinical settings which do not have the financial means to obtain a VHIT system. A less expensive diagnostic headband could be used during head impulses while the examiner observes the presence or absence of overt saccades [25]. 

### 4.5. Limitations

Testing was not randomized. SHIMP was always tested after HIMP since these tests were part of a whole testing day. However, if more coverts were produced during the second test (SHIMP) due to a learning effect, it would only underestimate the significant decrease of covert saccades in SHIMP. Moreover, previous studies with BV patients and healthy subjects showed no difference in covert saccades and/or VOR gain when tested repeatedly [11,16]. Therefore, it can be expected that randomization would not have significantly influenced the study. 

## 5. Conclusions

To our knowledge, this is the largest study population on SHIMP testing in BV patients. Covert saccades and VOR gains were significantly reduced during SHIMP compared to HIMP. However, the clinical relevance of these statistically significant differences is small, and both paradigms detect BV in the vast majority of patients. 

## Figures and Tables

**Figure 1 jcm-11-02444-f001:**
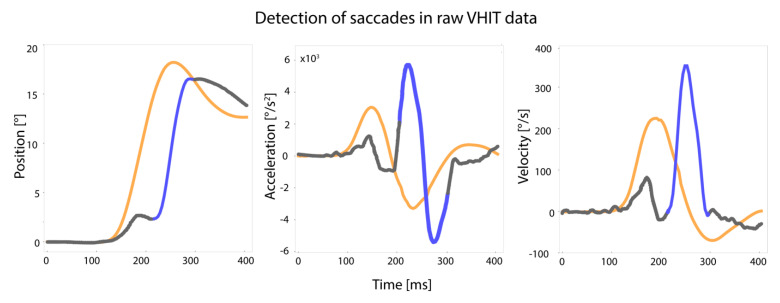
Detection of saccades in VHIT traces based on position, acceleration, and velocity of eye movement. The orange line illustrates the head impulse, the grey line represents the eye movement, and the blue line represents the saccade as included in the analysis. Raw data were exported from the Otometrics system (head and eye velocity traces). Position and acceleration data were calculated from these data. All traces were checked on artefacts and excluded if necessary. Saccades were extracted from these artefact-free traces using a custom-made algorithm. All saccades were verified by visual inspection. Definitions of artefacts and saccades are described in Section 2.3.

**Figure 2 jcm-11-02444-f002:**
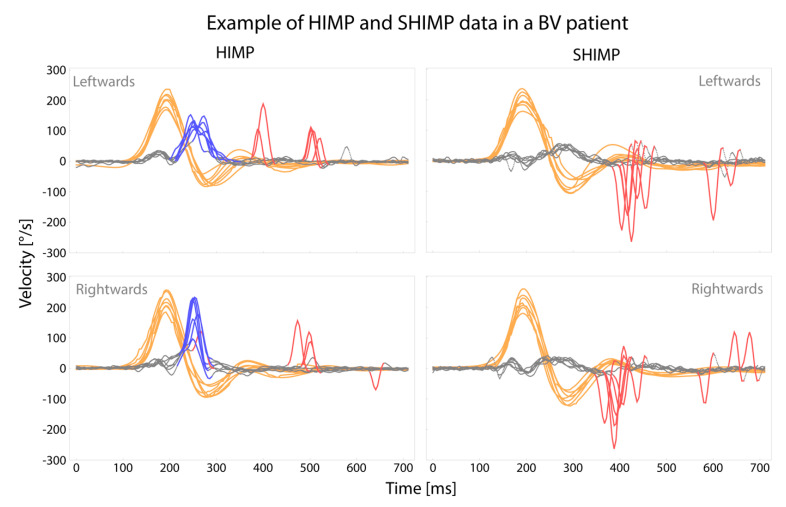
Raw eye and head movement data of one BV patient (patient 18), obtained by HIMP and SHIMP during two consecutive VHIT trials. Grey lines represent eye movements, orange lines represent head movements, blue lines represent covert saccades, and red lines represent overt saccades.

**Table 1 jcm-11-02444-t001:** Diagnosis of BV using HIMP and SHIMP (1a), and agreement between both paradigms (1b).

1a. Diagnosis According to VHIT Results (*n* = 92)	HIMP (Cut-Off < 0.6)	SHIMP (Cut-Off < 0.6)	SHIMP (Cut-Off < 0.5)
**Bilateral vestibulopathy**			
VOR gain <0.6 on both sides	64	70	65
**No bilateral vestibulopathy**			
VOR gain >0.6 on both sides	10	9	14
VOR gain >0.6 on one side	18	13	13
**1b. Agreement on the Diagnosis of BV between HIMP and SHIMP**			
HIMP (cut-off < 0.6) and SHIMP (cut-off < 0.6)	93%		
HIMP (cut-off < 0.6) and SHIMP (cut-off < 0.5)	97%		

## Data Availability

The data presented in this study are available on request from the corresponding author.

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
