# Peer review of "Suppression Head Impulse Test (SHIMP) versus Head Impulse Test (HIMP) When Diagnosing Bilateral Vestibulopathy"

_jcm, 2022, doi:10.3390/jcm11092444_

Round 1
Reviewer 1 Report
The authors studied the effects of elimination of covert saccades during SHIMP paradigm in the diagnosis of bilateral vestibulopathy (BV). For this purpose, they compared the outcomes of SHIMP and HIMP paradigms in a large group of 98 patients with BV. They found statistically significant fewer covert saccades and h-VOR gain with SHIMP as compared with HIMP protocol. SHIMP detects BV better than HIMP in patients previously diagnosed by caloric testing and/or rotatory chair testing. Agreement between HIMP and SHIMP on the diagnosis of BV was found in 93% (h-VOR gain <0.6) and 97% (h-VOR gain <0.5). Nevertheless, the clinical significance of these differences is small as both paradigms detect BV in most patients.
The manuscript is sound, well-written, and well-structured. The title reflects properly the subject of the manuscript. The abstract summarizes accurately the results of the research. The introduction is clear and concise, and the aims are well stated. The design is appropriate. The saccade detection and VOR gain calculation methods are very well described as is the statistical analysis. The results are presented clearly, and the table is not redundant. The discussion is very well organized, and limitations are argued. The conclusions are supported by the data. Important articles are included in the references.
Author Response
We appreciate you investing time to review our manuscript. Thank you so much for your on-point summary, and your very positive feedback.
Reviewer 2 Report
This is a nice article that compares the results obtained using the HIMP and SHIMP paradigms in a group of patients with bilateral vestibular hypofunction. The results obtained, in the terms of gains and the presence of saccades are analyzed, as well as the degree of concordance between both tests.
This reviewer considers that it is an article that provides great solidity in its results due to the high number of patients included.
The minor considerations that should be noted are:
- In the first sentence of the introduction (lines 33 and 34): I would respect the first description of the Head impulse test carried out by Halmagyi and Curthoys and would include the bibliographic citation.
- In line 38, replace “contralateral” with “opposite”
- Lines 40-41: “loss of gaze” should include “fixation”: loss of gaze fixation.
- Line 42: Not in all cases the presence of saccades represents vestibular hypofunction. Saccades may appear in patients with normal vestibular function (Matiño-Soler E, Esteller-More E, Martin-Sanchez JC, Martinez-Sanchez JM, Perez-Fernandez N. Normative data on angular vestibulo-ocular responses in the yaw axis measured using the video head impulse test. Otol Neurotol. 2015 Mar;36(3):466-71).
- Line 64: replace contralateral with “opposite”
- Line 71: Although it is briefly mentioned later, perhaps the influence of “predictability” on saccade latency in the SHIMP paradigm should be commented on here.
- Results. Although the sample was very heterogeneous… Were there differences in the presence of saccades depending on the aetiology?
Author Response
First of all, thank you for investing your time to review our manuscript. Your comments were very helpful. Please let us know if we did not interpret your feedback correctly. Please see the attachment for replies to all comments separately. Thanks
